# SPD: Sync-Point Drop for Efficient Tensor Parallelism of Large Language Models

**Han-Byul Kim** [1]  **Duc Hoang** [1]  **Arnav Kundu** [1]  **Mohammad Samragh** [1]  **Minsik Cho** [1]

## Abstract

With the rapid expansion in the scale of large language models (LLMs), enabling efficient distributed inference across multiple computing units has become increasingly critical. However, communication overheads from popular distributed inference techniques such as Tensor Parallelism pose a significant challenge to achieve scalability and low latency. Therefore, we introduce a novel optimization technique, Sync-Point Drop (SPD), to reduce communication overheads in tensor parallelism by selectively dropping synchronization on attention outputs. In detail, we first propose a block design that allows execution to proceed without communication through SPD. Second, we apply different SPD strategies to attention blocks based on their sensitivity to the model accuracy. The proposed methods effectively alleviate communication bottlenecks while minimizing accuracy degradation during LLM inference, offering a scalable solution for diverse distributed environments: SPD offered about 20% overall inference latency reduction with $< 1\%$ accuracy regression for LLaMA2-70B inference over 8 GPUs.

## 1. Introduction

Large Language Models (LLMs) (Gunter et al., 2024; Brown et al., 2020; Bubeck et al., 2023; Touvron et al., 2023a;b; Zhang et al., 2022; Penedo et al., 2023; Jiang et al., 2023) have revolutionized the field of natural language processing (NLP), driving significant advancements in a wide range of applications. Their ability to understand and generate human-like text has opened new possibilities for both research and practical uses. However, as these models grow in size and complexity, optimizing their performance becomes a crucial challenge, particularly in terms of latency.

A proven approach to achieving low latency is to run LLM inference in distributed computing environments, notably using Tensor Parallelism (TP) (Shoeybi et al., 2019). By sharding tensor operations into separate tracks or blocks and processing them simultaneously on parallel devices, TP significantly reduces inference time while maintaining accuracy.

However, to maintain mathematical parity with single-device inference, TP requires collective communication—often referred to as sync-points—throughout the model. These sync-points serve as communication barriers across all parallel devices to synchronize hidden representation tensors, as shown in Figure 1a. Because of this communicative nature, the overhead of sync-points depends on hardware constraints—such as the interconnects between devices and the network connections between nodes—which can become a bottleneck during execution. As LLMs grow in size, one must use more compute devices, which in turn increases the number of sync-points and further worsens inference latency. Therefore, in any distributed system, optimizing sync-points would greatly improve overall system performance.

To tackle this important issue, we propose **Sync-Point Drop** (SPD) a simple yet novel optimization technique with broad applications for LLM systems. Unlike existing works which tried to improve the communication process itself (NVIDIA, 2019; Jeaugey, 2019; Cheng et al., 2023) on system-level, SPD directly removes sync-point in the self-attention output (as in Figure 1b) within the target budget. To enable SPD directly on decoder block, we first introduce a block design for SPD that minimizes negative effects resulting from reduced communication (see Figure 3). Second, we apply SPD strategies differently to each blocks based on communication sensitivity, which we defined as the relative impact on downstream performance when all communications are dropped up to that point (see Figure 4). Our experimental results show effective possibility of latency improvement with minimizing the accuracy degradation throughout diverse sizes of models. In summary, our contributions are:

- We propose novel block designs for SPD that minimize

---

[1]Apple. Correspondence to: Han-Byul Kim <han-byul@apple.com>.

*Proceedings of the $42^{nd}$ International Conference on Machine Learning*, Vancouver, Canada. PMLR 267, 2025. Copyright 2025 by the author(s).

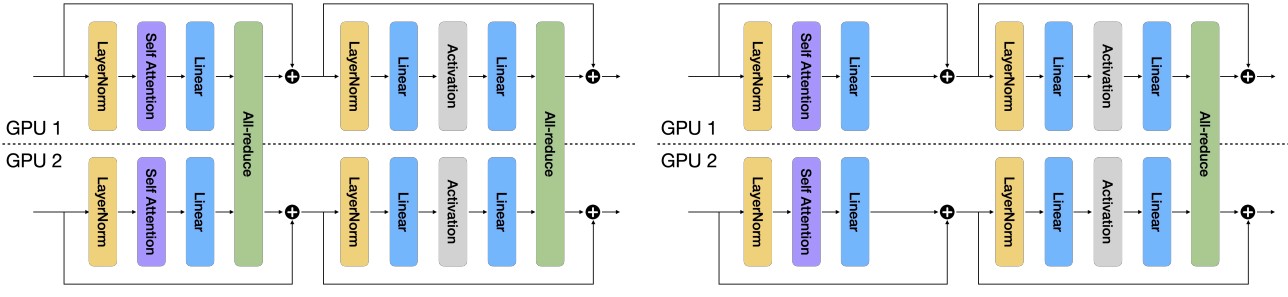

(a) With fully synchronous tensor parallelism.      (b) With the elimination of the sync-point in the attention output.

*Figure 1.* Tensor parallelism applied on transformer decoder block (in 2-GPUs distributed inference case).

information loss from lack of communication.

- We identify the sensitivity of each block within the model and classify them into three distinct categories, allowing for the application of tailored optimization strategies to each group based on their characteristics.

- Empirical results on various datasets and models show that proposed SPD with optimization strategies can offer better accuracy/latency trade-off by enabling a scalable solution for distributed environments and minimizing quality loss for overall communication budgets.

## 2. Related Works

The inefficiency of LLMs, which emerged with a significant impact on computation and memory, has led to a large demand for optimization techniques. Model-level optimization has gained attention for opportunities within the vast redundancies of LLMs. Quantization (Frantar et al., 2023; Xiao et al., 2023; Lin et al., 2024; Shao et al., 2024; Chee et al., 2023; Ashkboos et al., 2024) reduces the precision of model parameters, allowing for faster operations with minimal impact on performance. In particular, (Agrawal et al., 2024; Dong et al., 2024) are intended to optimize communication performance with low-bit expressions. Pruning (Frantar & Alistarh, 2023; Sun et al., 2024; Liu et al., 2023; Xia et al., 2024) eliminates less critical parameters or neurons from the model, thereby reducing its size and computational complexity. In the aggressive scale, block skipping (Xia et al., 2024; Song et al., 2024), which involves bypassing certain blocks, further enhances efficiency by eliminating the operations of a block. These approaches focus on compressive and computational effects which makes the model suitable for real-time and resource-constrained environments.

Following the underlying mechanism of model deployment, beyond model-level optimization, system-level optimizations (Shoeybi et al., 2019; Huang et al., 2019; Zhao et al., 2023; Aminabadi et al., 2022; Kwon et al., 2023) are explored. Different from model-level approach, system-level optimization does not change any numerical values of a model. One of the distributed deployment techniques, tensor parallelism (Shoeybi et al., 2019), enables fast serving of a model by parallel execution of a block into multiple devices. However, this technique requires large communication overheads between devices to keep the numeric precision of execution flow. Considering the communication bottleneck of tensor parallelism, existing works also focus on improving the communication operation itself systematically, including ring-topology all-reduce (NVIDIA, 2019) and tree-topology all-reduce (Jeaugey, 2019). Specifically for large models, ATP (Cheng et al., 2023) improves training efficiency by dynamically choosing the parallel strategy.

In this paper, we leverage optimization benefits in model-level from the system perspective (enabling SPD in the system while minimizing accuracy degradation in the model).

## 3. Preliminary: Tensor Parallelism in LLMs

Tensor Parallelism (TP) (Shoeybi et al., 2019) is a systematic computing technique on a distributed environment used to accelerate large-scale language models. This is realized by partitioning individual weight tensors of a model across multiple devices. Instead of replicating the entire model across GPUs (as in data parallelism), TP divides each block's computation across multiple devices (as in Figure 1a), enabling the model to handle larger tensors that would otherwise exceed the memory capacity of a single GPU. This approach significantly improves the scalability and efficiency in both training and inference, particularly in LLMs. However, the realization of effective TP requires collective communication (*all-reduce* in Figure 1) between devices to synchronize and exchange partial computations. The communication latency is typically decided by network bandwidth between devices. The lower the bandwidth, the more the parallel system gets bottleneck originated from the sync-point. To resolve the bottleneck of distributed inference, our proposed optimization technique, SPD, simply eliminates the communication within each decoder block (as in Figure 1b).

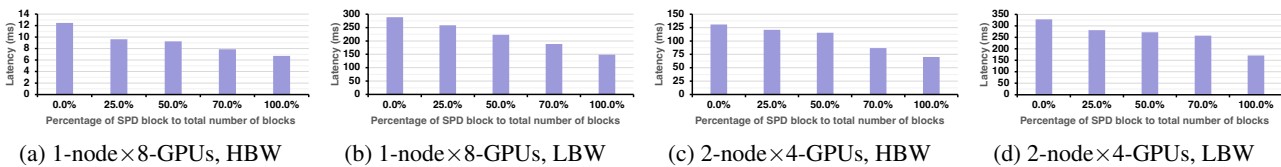

(a) 1-node×8-GPUs, HBW  (b) 1-node×8-GPUs, LBW  (c) 2-node×4-GPUs, HBW  (d) 2-node×4-GPUs, LBW

*Figure 2.* Data transfer latency of LLaMA2-70B distributed inference with SPD on different system settings of NVIDIA A100-80G GPU node. 'HBW' represents high bandwidth setting and 'LBW' represents low bandwidth setting for device interconnect. Input consist of batch size of 1 and sequence length of 128 is used.

Figure 2 shows the data transfer latency of *all-reduce* incurred by GPU kernel. The metrics are measured on LLaMA2-70B distributed inference in different system settings with diverse levels of SPD across the model blocks. The measurement is conducted under both high bandwidth (HBW; 300GB/s interconnect) and low bandwidth (LBW; 10GB/s interconnect) device interconnect (Detailed settings about connection bandwidth are explained in Section 5.1). Applying SPD at 100%, which halves the number of sync points in the entire model, significantly reduces data transfer latency (over 46% reduction in all system settings), resulting in substantial overall model latency improvement across diverse system configurations (as in Figure 7c). This highlights the importance of addressing communication bottlenecks for efficient distributed inference. However, reducing sync points to minimize latency may lead to disrupted numerical parity, which does not always guarantee non-degraded accuracy. To address this, we propose novel block design and techniques combined with SPD that alleviate communication bottlenecks while minimizing quality loss across various SPD budget cases, providing a scalable solution for distributed inference systems.

# 4. Sync-Point Drop

As an efficient method to improve distributed inference performance, Sync-Point Drop (SPD) selectively removes the *all-reduce* communication operation after self attention output, as illustrated in Figure 1b. In this section, we discuss how to maintain high model quality with reduced communication overhead. First, we introduce a novel block structure design that serves as the foundation block for the non-communicating structure with minimal quality degradation. Second, we propose a strategy of applying SPD in a block-wise manner which achieves lower latency with minimal accuracy loss.

## 4.1. Block Design

If the synchronization of the self attention output is skipped, the attention output will diverge across multiple devices. Therefore, SPD requires two architecture changes in the transformer block, the MLP input and output in a way that information loss from SPD can be minimized. Also, the

complexity of such changes increases if the output projection layers in self attention include a bias.

### 4.1.1. BLOCK WITHOUT BIAS IN LINEAR LAYER

Figure 3a shows the SPD block design used without bias in linear layer. The most essential objective of block design is constructing the combination of connections which gives least numerical difference between TP and SPD.

**MLP input** The sync-point enables each parallel device to capture the attention output from all the other devices. However, when the outputs from other devices are unavailable by elimination of sync-point, the only information the device $(i)$ can utilize is its own attention output $(Y_i)$. Therefore, to minimize the numerical difference compared to the case of all information available, residual connection $(X)$ and attention output of own device $(Y_i)$ are added and fed into MLP input $(X + Y_i)$.

**MLP output** When the sync-point exists after attention output, the MLP input is utilized as a residual connection added to the MLP output. However, dropping the sync-point yields incomplete MLP input $(X + Y_i)$ with lack of attention outputs from other devices. The desired block output is a combination of block input $(X)$, attention output from all devices $(\sum_i Y_i)$, and MLP output from all devices $(\sum_i Z_i)$. Therefore, we disassemble the original residual connection to block input residual $(X)$ and attention output residual from a device $(Y_i)$. Then, $Y_i$ forms a new type of residual connection which is added before the sync operation. $X$ is added on the same point as the original connection, after the sync operation which finally leads to a complete form of output $(X + \sum_i Y_i + \sum_i Z_i)$.

### 4.1.2. BLOCK WITH BIAS IN LINEAR LAYER

In TP, each of the linear layers in self attention part of a block is parallelized in a different manner. The linear layers before self attention operation (query, key, and value projection) are divided in a column-wise manner which enables the bias divided along the same dimension. However, the linear layer after self attention operation (output projection) is parallelized in an orthogonal way, row-wise manner. The bias, a vector along the column dimension, therefore, can not be divided in the direction of the row. This requires a

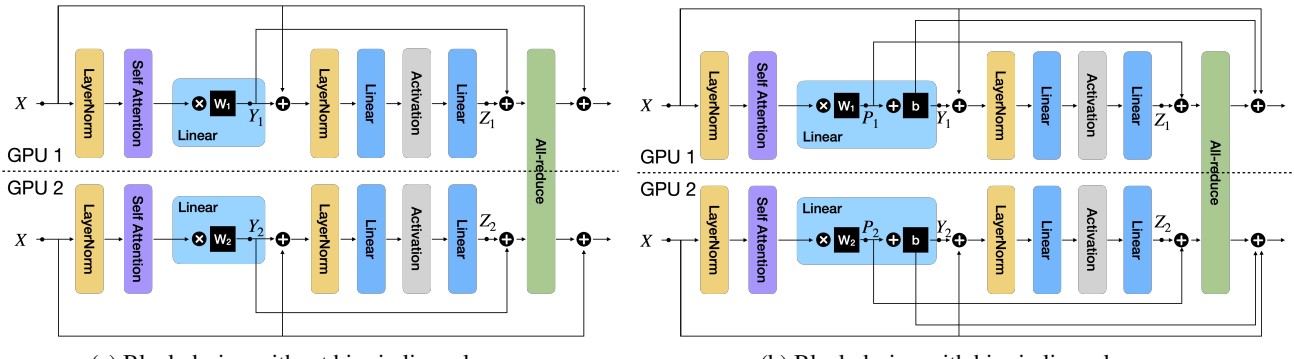

(a) Block design without bias in linear layer.      (b) Block design with bias in linear layer.

*Figure 3.* Decoder block structure with sync-point drop (in 2-GPUs distributed inference case). '$W_i$' and '$b$' represent weight and bias of linear layer on each device ($i$). '$X$', '$Y_i$', '$Z_i$' and '$P_i$' denotes a hidden representation of each device ($i$) on '·' in the figure.

new mechanism of the bias application on MLP input and output as shown in Figure 3b.

**MLP input** Different from the case with bias (Section 4.1.1), the indecomposable bias term ($b$) is included after weight multiplication. Following the essential objective, the least error in the MLP input compared to the result of TP, we use the partial weight multiplication result with the addition of bias ($Y_i = P_i + b$) and input residual connection ($X$) as MLP input ($X + P_i + b$).

**MLP output** Following Section 4.1.1, the original residual connection is disassembled to block input residual ($X$) and attention output residual ($Y_i$) from a device. Due to the existence of bias, we further disassemble $Y_i$ to the result of the partial weight multiplication ($P_i$) and the bias ($b$). To make the bias not affected by communication, we place the bias residual add after the sync operation while adding the partial weight multiplication result before the sync operation. Finally, in a device, this makes the bias residual be added once on MLP output while the parallelized weight multiplication results form a complete state through collective communication ($X + \sum_i P_i + b + \sum_i Z_i$).

### 4.2. Sync-Point Drop based on Block-wise Sensitivity

While the lack of communication incurs numerical disparity across all parallel devices, such disparity in different blocks will impact the model accuracy differently. In this section, we introduce a multi-tiered block-wise approach to minimize the overall accuracy loss based on the SPD-specific per-layer sensitivity.

First, in Section 4.2.1, we categorize transformer blocks based on their sensitivity to SPD: in-sensitive blocks (ISB), sensitive blocks (SB), and extremely sensitive blocks (ESB). Based on the classification result, before applying SPD, we perform individual preprocessing steps (Section 4.2.2, 4.2.3, and 4.2.4). This specific strategy allows us to minimize accuracy degradation from SPD thereby enabling a

better balance between model performance and quality on deployment.

#### 4.2.1. BLOCK-WISE SYNC SENSITIVITY IDENTIFICATION

To identify the sensitivity of a block to SPD, we utilize the perplexity metric by measuring the relative impact of a block to performance (the difference from application between TP block and SPD block in Figure 4) as sensitivity measurement. For example, when we measure the sensitivity of $i$-th block to SPD, we apply SPD to all blocks starting from the $\{i + 1\}$-th block to the final block and measure the perplexity, while leaving the $i$-th block unchanged (TP block). We then measure the perplexity by additionally modifying the system setting of $i$-th block to SPD. The difference in perplexity before and after applying SPD to $i$-th block is used as a measure of sensitivity. In this measurement, we use calibration data obtained by sampling a small portion of the large training dataset. By progressive replacement of TP block to SPD block and measurement of quality degradation as relative perplexity difference, we can compare the sensitivity between blocks in the entire model and classify the blocks into three sensitivity categories (ISB, SB, and ESB).

Algorithm 1 shows the overall process of applying SPD in a multi-tiered block-wise approach with measured sync sensitivity. Based on the sync sensitivity value of blocks ($S$), we rank the blocks in an ascending order ($B$). According to the predetermined ranking of the sensitivity, SPD is applied within the target number of blocks to optimize ($N_{spd}$). In the sequence, the processing of a block is classified into three sensitivity categories based on predefined threshold criteria ($\tau_1$ and $\tau_2$). This classification allows us to apply separate approaches aimed at minimizing quality degradation according to the identified groups. In the following sections, we introduce the individual strategies applied to three categories of the blocks.



**Figure 4.** Sync sensitivity identification of a decoder block (measuring the sensitivity of $i$-th block).

---

**Algorithm 1** Sync-point drop based on sensitivity

---

1: SPD: SYNC-POINT DROP
2: B2B: BLOCK-TO-BLOCK DISTILLATION
3: HG: ATTENTION HEAD GROUPING INITIALIZATION
4: $Block \leftarrow$ *list of all decoder blocks in model*
5: $S \leftarrow$ *list of sensitivity measurement*
6: $B \leftarrow$ *block index list in ascending order of $S$*
7: $N_{spd} \leftarrow$ *Target budget: the number of blocks to SPD*
8: $\tau_1, \tau_2 \leftarrow$ *sensitivity thresholds*
9: **for** $i = 0$ **to** $N_{spd} - 1$ **do**
10:    **if** $S[B[i]] \leq \tau_1$ **then** ▷ Categorize as ISB: Section 4.2.2
11:        $Block[B[i]] \leftarrow$ SPD($Block[B[i]]$)
12:    **else if** $S[B[i]] \leq \tau_2$ **then** ▷ Categorize as SB: Section 4.2.3
13:        $Block[B[i]] \leftarrow$ SPD(B2B($Block[B[i]]$))
14:    **else** ▷ Categorize as ESB: Section 4.2.4
15:        $Block[B[i]] \leftarrow$ SPD(B2B(HG($Block[B[i]]$)))
16:    **end if**
17: **end for**

---

#### 4.2.2. IN-SENSITIVE BLOCKS: ZERO-SHOT DROPPING

ISBs show minimal quality degradation with SPD. Therefore, within the targeted budget of communication optimization ($N_{spd}$), we drop the sync-point of these blocks, prioritized over other types of blocks, in a zero-shot manner. Note that zero-shot dropping can give a significant amount of benefit with sensitivity identification. As shown in Section 5, in every model, zero-shot dropping can obtain at least 44% of blocks as SPD blocks with little sacrifice of accuracy.

#### 4.2.3. SENSITIVE BLOCKS: SPD AWARE BLOCK-TO-BLOCK DISTILLATION

SBs exhibit larger effects on quality degradation compared to ISBs. To further achieve the optimization objectives and recover the associated performance degradation in SB, we obtain SPD aware parameters by adopting block-to-block distillation. Block-to-block distillation is a low-cost fine-tuning method that involves training only the specific SB with SPD setting. We set the teacher block as TP block and the student block as the SPD block. For the data used in tuning, we utilize calibration data used in the sensitivity identification step in Section 4.2.1. This data passes through consecutive TP blocks of the model by the block in which distillation will be conducted. Then, we utilize the hidden representation of the block's input where distillation will be performed. Following the fine-tuning objective in Equation 1, we forward the hidden representation ($x$) to each teacher and student block and apply outputs to mean squared error (MSE) loss. Note that the parameter of SPD block ($\theta_{spd}$) is initialized from the parameter of TP block ($\theta$). Since SPD and TP are execution methods within the system, they

originally use the same model parameters. However, to obtain the special weights aware of eliminated communication, parameters for SPD are newly initialized from the original and used separately.

$$\underset{\theta_{spd}}{\mathrm{argmin}}\ \mathrm{MSE}(\mathrm{SPD}(\theta_{spd}, x), \mathrm{TP}(\theta, x)) \qquad (1)$$

#### 4.2.4. EXTREMELY SENSITIVE SLOCKS: SPD AWARE ATTENTION HEAD GROUPING INITIALIZATION

Beyond the recovery of block-to-block distillation on SBs, a few blocks show sharp quality degradation. We define these blocks as ESBs and introduce a novel SPD aware initialization before conducting block-to-block distillation. As the sync-points are removed, the model partitions located on each device are isolated from each other, preventing mutual access. This makes a decoder block as if it is a combination of parallel and independent mini decoder blocks. In this circumstance, a self attention fragment cannot access any MLP partitions in other parallel devices and also MLP partitions are unable to access self attention output in other parallel devices, resulting in inevitable information loss. To ensure that these parallel architectures operate as close as the original structure, it is important to make attention heads evenly distributed based on functionality following the sparse nature of head activated differently (Liu et al., 2023) and redundancy of head showing similar behaviors (Agarwal et al., 2024) on in-context. To reflect these in-context properties to out-context as much as possible, we utilize calibration data and obtain attention score ($\sigma$) as a metric of the head functionality.

**Head scattering** In the self attention, the set of the query

(Q), key (K) and value (V) associated with each head can be defined as $A = \{<Q_1, K_1, V_1>, <Q_2, K_2, V_2>, \cdots, <Q_N, K_N, V_N>\}$ where $N$ is the number of heads. The goal of head scattering is finding the set of heads showing the even distribution of attention score ($\sigma(Q_i, K_i)$) across the parallel devices. By defining a set of heads to be placed in a device as $A_i$ where $A_i \subset A$ and $n(A_i) = N/number\_of\_devices$, the objective of head scattering is defined in Equation 2. We achieve the objective of finding an even distribution based on head functionality by maximizing the sum of distances on the clustering algorithm which originally utilized the opposite metric. For the distance, attention scores of each sequence as a high dimension vector are utilized with euclidean distance ($d$).

$$\underset{A_i}{\arg\max} \sum_{j=1}^{n(A_i)} \sum_{k=j+1}^{n(A_i)} d\Big(\sigma(Q_{A_{i,j}}, K_{A_{i,j}}), \\ \sigma(Q_{A_{i,k}}, K_{A_{i,k}})\Big), \quad \text{where} \quad A_i \subset A \tag{2}$$

**MLP matching** After getting the scattered clusters of attention heads, matching $A_i$ with proper MLP partition (sharded MLP operation in a parallel device) should be conducted to search for complete parallel independent architecture that operate close to the original structure. We found that the norm of MLP output before adding residual connection is a well-fit indicator. The MLP output norm is small compared to the residual connection norm (Liu et al., 2023). A large block output norm implies that the block contributes well when combined with the residual. Therefore, we compare the norm of all the matching combinations and pick the best maximum case as the matching result. By defining the MLP partition of a device as $MLP_m$ and a matching combination as $MC$ and its universal set (all combinations) as $MC_{all}$, the objective of MLP matching is defined as Equation 3.

$$\underset{MC}{\arg\max} \sum_{<A_i, MLP_m>}^{MC} Norm(MLP_m(A_i)), \\ \text{where} \quad MC \in MC_{all} \tag{3}$$

After determining the optimal $A_i$ and $MC$, the hidden representation of each head should be physically located on the device designated by $MC$. Figure 5 illustrates the example of SPD with best $A_i$ and $MC$. To align the assignment with the static behavior of the system in SPD, we reorder the columns of the query, key, and value linear layer weights ($W_Q, W_K, W_V$) based on their head-specific partitions ($W_{Qh}, W_{Kh}, W_{Vh}$, where $h$ denotes the head index). Similarly, the row order of output linear layer weight ($W_O$) based on head partitions ($W_{Oh}$) is reordered. This reordering ensures that the hidden representations are distributed in the order of $MC$, allowing the heads in $A_i$ to

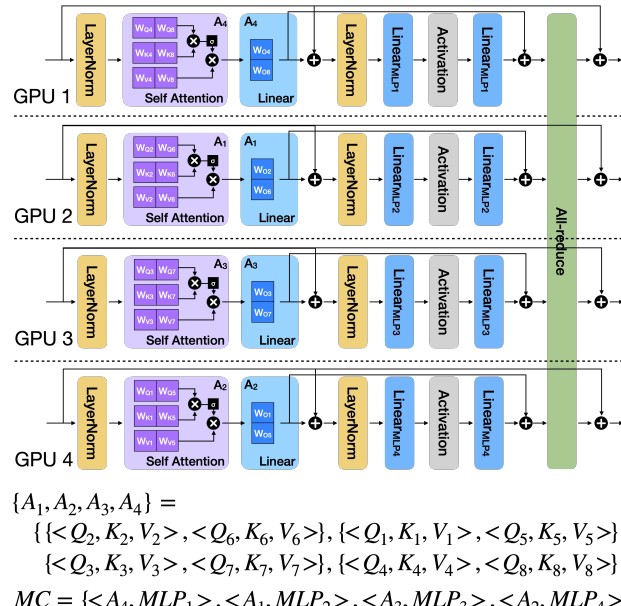

$\{A_1, A_2, A_3, A_4\} = $
$\{\{<Q_2, K_2, V_2>, <Q_6, K_6, V_6>\}, \{<Q_1, K_1, V_1>, <Q_5, K_5, V_5>\},$
$\{<Q_3, K_3, V_3>, <Q_7, K_7, V_7>\}, \{<Q_4, K_4, V_4>, <Q_8, K_8, V_8>\}\}$
$MC = \{<A_4, MLP_1>, <A_1, MLP_2>, <A_3, MLP_3>, <A_2, MLP_4>\}$

*Figure 5.* SPD block in case having 8-heads on 4-GPUs parallel with given head subset ($A_i$) and matching combination ($MC$).

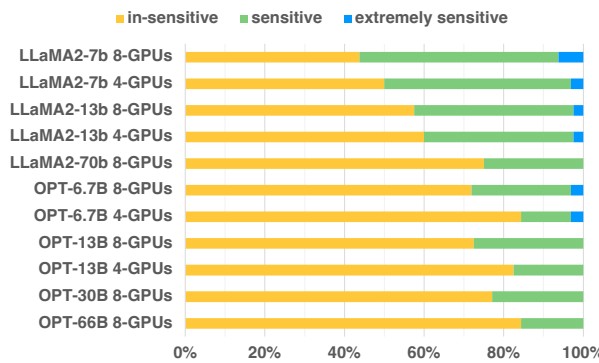

*Figure 6.* Block-wise sync sensitivity identification result for LLaMA2 and OPT models over 8-GPUs and 4-GPUs.

reside on the same parallel device. As a result, a group of scattered heads subset and MLP partition is assigned to a single device, working as SPD aware initialization. Applying block-to-block distillation after head grouping further enhances accuracy recovery in the ESBs.

## 5. Experiments

### 5.1. Setup

**Models** We conduct experiments on LLaMA2 (7B, 13B, and 70B) (Touvron et al., 2023b) and OPT (6.7B, 13B, 30B, and 66B) (Zhang et al., 2022). We apply 8-GPUs and 4-GPUs distributed inference for all the models except LLaMA2-70B, OPT-30B, and 66B which apply 8-GPUs setting only.

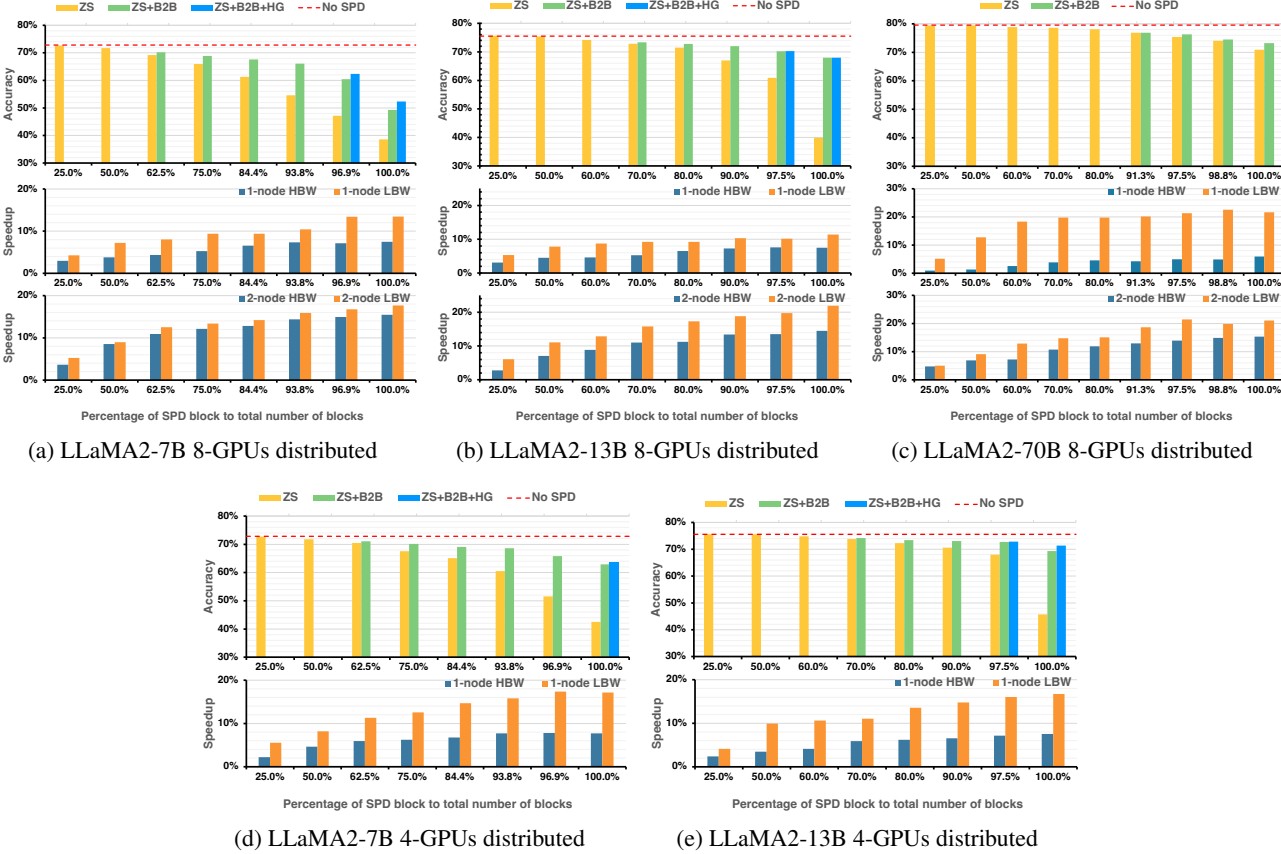

(a) LLaMA2-7B 8-GPUs distributed     (b) LLaMA2-13B 8-GPUs distributed     (c) LLaMA2-70B 8-GPUs distributed

(d) LLaMA2-7B 4-GPUs distributed     (e) LLaMA2-13B 4-GPUs distributed

*Figure 7.* LLaMA2 distributed inference average accuracy on zero-shot tasks (top) and normalized speedup of time to first token (bottom). Speedup is normalized based on latency of 0% (No SPD state which consists of TP blocks in entire model) in each distributed inference setting. 'ZS' represents applying zero-shot dropping to all blocks. 'ZS+B2B' represents applying zero-shot dropping on ISBs and block-to-block distillation to the other remaining SBs and ESBs. 'ZS+B2B+HG' is applying zero-shot dropping on ISBs and block-to-block distillation to SBs and block-to-block distillation with head grouping initialization to the other remaining ESBs. 'HBW' represents high GPU interconnect bandwidth of 300GB/s setting and 'LBW' represents low GPU interconnect bandwidth of 10GB/s setting.

**Calibration data** From WikiText2 (Merity et al., 2016) training dataset, randomly selected 128-samples consisting of tokens with a sequence length of 2048 are used by following existing work (Shao et al., 2024). Each sample of calibration data is utilized as a mini batch for distillation.

**Evaluation data** We evaluate the accuracy of our optimization method to zero-shot tasks (ARC (Clark et al., 2018), HellaSwag (Zellers et al., 2019), LAMBADA (Paperno et al., 2016), PIQA (Bisk et al., 2020), SciQ (Welbl et al., 2017), and WinoGrande (Sakaguchi et al., 2020)) by averaging all the results and MMLU tasks (Hendrycks et al., 2021).

**Hyper-parameter setting** For all models except larger models (LLaMA2-70B, OPT-30B, and OPT-66B), we use $\tau_1$ as 0.05 and $\tau_2$ as 10. For larger models, we use $\tau_1$ as 0.02 and $\tau_2$ as 10. In block-to-block distillation on SBs and ISBs, the learning rate is used as $5 \times 10^{-5}$ for LLaMA2 and $1 \times 10^{-6}$ for OPT. 10-epochs distillation is conducted with

each 1-epoch utilizing whole 128-samples of calibration data.

**Environment setting** The accuracy and latency of all our experiments are measured on nodes with Nvidia A100-80G GPU node under high (300GB/s) and low (10GB/s) bandwidth GPU interconnect setups following (Cho et al., 2024). The low bandwidth interconnect is established by turning off the high-speed CUDA-direct link (NVIDIA, 2019). For 2-node of 8-GPUs distributed cases, each node consists of 4-GPUs with 50GB/s interconnect between nodes.

### 5.2. Sensitivity Identification

Figure 6 shows the block-wise sync sensitivity identification result of the blocks in LLaMA2 and OPT models. For all models, the percentage of ISBs (yellow bar) indicates that the same amount of blocks can be used as SPD with an ignorable accuracy drop (less than about 1% on zero-shot tasks). This can be achieved in the zero-shot manner (de-

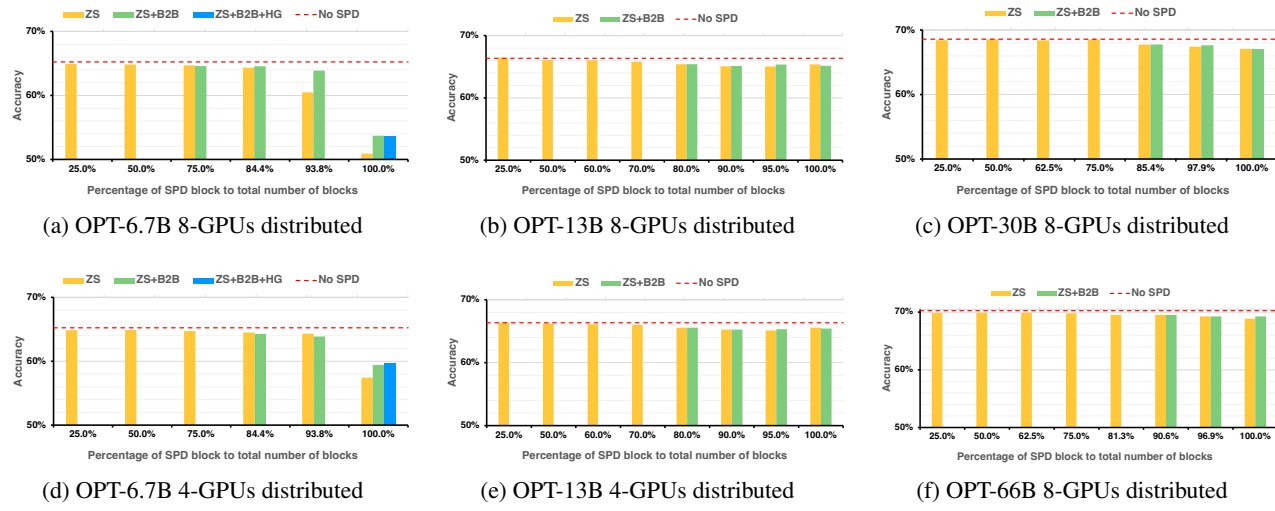

*Figure 8.* OPT distributed inference average accuracy on zero-shot tasks (Notations are same as in Figure 7).

tailed results are described in Section 5.3). The percentage of ISBs increases when the model size gets larger (75% in LLaMA2-70B 8-GPUs and 84% in OPT-66B 8-GPUs). Overall, LLaMA2 models show higher sensitivity compared to OPT models. LLaMA2-7B 8-GPUs model is available with a zero-shot drop of 44% while entire OPT models are available with dropping 70% of blocks. ESBs are shown only in smaller models (LLaMA2-7B, 13B, and OPT-6.7B) with one or two blocks.

### 5.3. Sensitivity based Sync-Point Drop

Figure 7 shows the SPD results of LLaMA2 models on zero-shot tasks. After the amount of target SPD blocks exceeds in-sensitive boundary, zero-shot dropping (ZS) shows large accuracy drop (over 1%) in all models and system settings. Block-to-block distillation with ZS (ZS+B2B) successfully recovers large amount of accuracy degradation in SB region, especially giving larger amount on smaller models (+28% on 13B 8-GPUs of Figure 7b and +20% on 7B 4-GPUs of Figure 7d on 100% SPD). Furthermore, smaller models having ESBs show further accuracy recovery from B2B (+3% on 7B 8-GPUs of Figure 7a and +2% on 13B 4-GPUs of Figure 7e on 100% SPD) with adding head grouping initialization (ZS+B2B+HG). Similar tendencies are appeared on MMLU results in Appendix A.

SPD brings benefits of latency improvement from the elimination of sync point while the multi-tiered block-wise approach recovers accuracy degradation. In 1-node cases, overall, the lower the device interconnect bandwidth, the larger the speedup SPD achieves. For LLaMA2-70B with LBW in Figure 7c, 70% SPD offers about 19.7% speedup while only sacrificing 0.94% accuracy. Note that this result is from simple zero-shot dropping from ISBs. In 2-node cases (Figure 7a, 7b and 7c) where the distributed system

has connection between the nodes, both HBW and LBW interconnect setups show significant amount of latency improvement. For all models with 2-node system, SPD shows over 10% speedup on both HBW and LBW of SPD percentage over 70%. In LLaMA2-13B (Figure 7b) and 70B (Figure 7c), speedup over 20% can be achieved on the extreme level of SPD (nearly 100%) with LBW setting.

Figure 8 shows the SPD results of OPT models on zero-shot tasks. OPT models show less drop compared to LLaMA2 models possibly due to high redundancy (Liu et al., 2023; Agarwal et al., 2024). Models except 6.7B show a maximum 1.3% degradation regardless of the sensitivity of the block. Therefore, results in OPT with ZS+B2B show small improvements since they already have less drop only with ZS. However, in OPT-6.7B (Figure 8a and 8d), when the drop occurs in ZS by increasing the percentage of SPD block, ZS+B2B and ZS+B2B+HG give recovered accuracy (+2.8% in 8-GPUs of Figure 8a and +2% in 4-GPUs of Figure 8d on 100% SPD).

Overall the proposed SPD effectively alleviates sync-point bottleneck while minimizing accuracy degradation. This shows that SPD gives both moderate optimization with no performance degradation and the better trade-off between larger optimization and performance leading to a scalable solution.

## 6. Conclusion

In this paper, we present Sync-Point Drop (SPD), a novel optimization technique that improves the latency of LLMs on distributed inference systems. By adopting a new block design and separated approaches based on block-wisely identified sensitivity for lack of sync-point, SPD enables efficient deployment across multiple computing units with

little compromising model performance. Our experiments show that SPD successfully alleviate communication overhead in tensor parallelism with minimum quality loss in all budgets, which enable scalable solution for distributed inference systems.

## Impact Statement

This paper presents work whose goal is to advance the field of Machine Learning. There are many potential societal consequences of our work, none which we feel must be specifically highlighted here.

## Acknowledgments

We thank Rasoul Shafipour, Raviteja Vemulapalli, Chenfan Sun, Iman Mirzadeh, Keivan Alizadeh Vahid, Devang Naik, Kulin Seth, and Brendan Langoulant for valuable discussions and comments.

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

## A. Sensitivity based sync-point drop

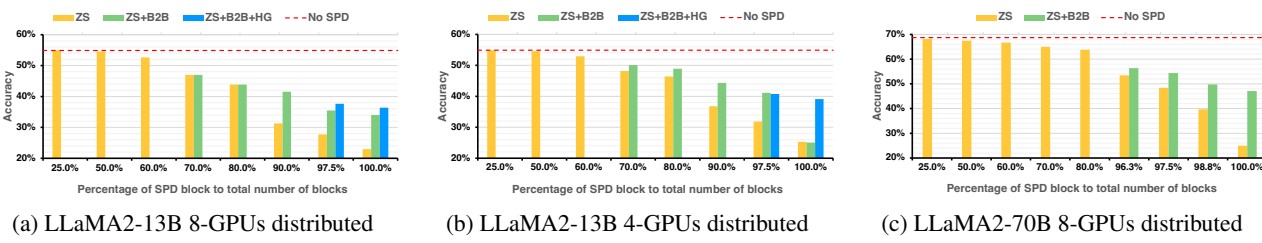

(a) LLaMA2-13B 8-GPUs distributed   (b) LLaMA2-13B 4-GPUs distributed   (c) LLaMA2-70B 8-GPUs distributed

*Figure 9.* LLaMA2 distributed inference accuracy on MMLU tasks (Notations are same as in Figure 7).

## B. Ablation study

### B.1. Effects of design choice in block design

*Table 1.* SPD MLP output design choice WikiText2 perplexity on block (SPD is only on 1st block of the model).

| Attention output residual add ($Y_i$) | PPL ($\downarrow$) | Bias residual add ($b$) | PPL ($\downarrow$) |
|---|---|---|---|
| LLaMA2-7B no SPD | 5.47 | OPT-6.7B no SPD | 10.86 |
| Before MLP all-reduce | 10.65 | Before MLP all-reduce | 332.60 |
| After MLP all-reduce | 177.69 | After MLP all-reduce | 13.07 |
| (a) Without bias in linear layer | | (b) With bias in linear layer | |

Section 4.1 shows that the tensor parallelism block system is not compatible with lack of communication and this makes several design choices on block structure. Table 1a and 1b show quality degradation per design choice on MLP output. Whether the targeted residual connections on each table use collective communication or not will be determined by the addition point (before and after MLP all-reduce). The results show that using collective communication on attention output residual (Table 1a) and not using it on bias (Table 1b) are the proper choice of residual addition point design selections as in Figure 3 which minimizes negative effect from SPD.

## C. Discussion

### C.1. Sensitivity identification

In Section 4.2.1, when measuring the sensitivity of a block to SPD, we apply SPD to consecutive blocks starting from the selected block and extending toward the last block. This approach allows us to evaluate the worst-case impact of SPD on the block's output while ensuring that its input activations remain numerically identical to the original, unaffected by prior SPD modifications. By analyzing the perplexity difference between a given block and the next block, we can isolate the effect of SPD at that specific layer, excluding any influence from subsequent layers. This method provides a fast yet precise and independent measurement of each block's sensitivity.

### C.2. Compatibility with other parallelism

SPD can be integrated with diverse settings to suit specific application contexts. In this section, we describe how SPD can be combined with such settings.

**Data parallelism** Data parallelism can be achieved with SPD by replicating an SPD model across distributed GPU environments. For example, in an 8-GPUs configuration, a 4-GPUs SPD model can be instantiated as a single replica, which is then duplicated across the remaining 4-GPUs. Consequently, two replicas operate in parallel, forming a data parallelism setup with SPD.

**Pipeline parallelism** Pipeline parallelism can be incorporated into SPD by applying it at the level of each parallelized decoder block, in which the attention and feed-forward network (FFN) components are already distributed across multiple GPUs. For instance, in an SPD configuration where a single decoder block is partitioned across 4-GPUs, pipeline parallelism

can be realized by further dividing the entire distributed blocks into two sequential groups. This results in a 4×2-GPUs system, where each of the four SPD branches is executed in a two-stage pipeline across a pair of GPUs, thereby enabling pipeline parallelism within SPD blocks.

**Hybrid parallelism** A pipeline-parallelized model can be instantiated as a replica to enable data parallelism. By replicating this pipeline-parallelized model, data parallelism can be applied on top of the pipeline setup. This approach ultimately facilitates the implementation of combined parallelism in conjunction with SPD.

### C.3. Impact of SPD on tensor parallelism compared to pipeline parallelism

In this work, we focus on optimizing tensor parallelism (TP) by addressing the communication bottlenecks that frequently arise in distributed inference systems. Although TP is widely adopted in practice due to its engineering simplicity and high compute utilization, pipeline parallelism (PP) is often considered a promising alternative, primarily due to its lower communication overhead. Nonetheless, TP remains the preferred strategy in many real-world distributed inference deployments, even under constrained interconnect bandwidth, due to its ability to more effectively utilize available compute resources.

While the relative reduction in communication volume achieved by SPD may appear modest (maximum 50% since SPD only targets the sync-point after self attention output), its impact on end-to-end latency is substantial. This is because the reduction in communication bottlenecks allows the remaining computational tasks, which are less affected by the system's bandwidth, to proceed more efficiently, ultimately improving the end-to-end performance. Therefore, TP optimized by SPD is more effective in deployment scenarios where compute resources need to be fully utilized across GPUs, as seen in prefill stage of large language model (Figure 7).

