# OpenReview forum: "SPD: Sync-Point Drop for Efficient Tensor Parallelism of Large Language Models"
_ICML.cc/2025/Conference — ICML 2025 poster_

### Official Review · Reviewer_J4A1 · 2025-03-06

**Overall Recommendation:** 4

**Summary:**

Sync-Point Drop (SPD) is an inference-time technique for reducing the latency of distributed inference with tensor parallelism. The paper suggests removing all-reduce operation between attention and feed-forward layers of transformer blocks. At the same time, some of the residual connections within the transformer block also need to be added/modified. They refer to this compound modification as SPD. The authors show that it is possible to apply it without significantly losing the accuracy of the model.

The authors start sync-point dropping from the later layers of the model. As they proceed with each block, they measure the perplexity of the resulting configuration. They compute differences between consecutive perplexity evaluations and take those as the sensitivity values of each individual block. They sort the blocks by sensitivity and assign them to three tears: insensitive, sensitive, and especially sensitive. They recognize that applying SPD as-is (zero-shot) to sensitive blocks degrades the performance, so they come up with two additional techniques based on attention head reordering (HG) and distillation (B2B) explicitly directed at them.

**Claims And Evidence:**

The paper presents well-supported claims related to Sync-Point Drop (SPD). The experimental results would benefit from including margin of error for measurements, especially for the measurements of latency.

The paper would benefit from quantifying the potential speed-up in set-ups that do not suffer communication bottlenecks. Detailed profiling of an inference step (before and after modifying the block) is missing.

**Essential References Not Discussed:**

The Section 2 of the paper gives a good overview of the research on improving the latency of comms-bound inference systems.

**Experimental Designs Or Analyses:**

The experimental setup in Section 5 is elaborated clearly. There are no obvious errors in the experiments or their analysis.

It remains unclear how much overlap between communication and computation the authors were able to achieve in their experimental setup.

The analysis would benefit from a more obvious illustration of the trade-off between latency and accuracy of the model.

**Methods And Evaluation Criteria:**

Measuring the model performance in terms of downstream task accuracy and latency ms is a reasonable approach.

An alternative to measuring latency could be a characterization of just how much communication-computation gap this approach can close in realistic scenarios.

The way sensitivity is measured looks arbitrary. Why don’t I measure sensitivity by applying SPD to individual blocks instead? Why don’t I start applying SPD to the blocks at the beginning of the model instead of applying it at the end?

**Other Comments Or Suggestions:**

Figure 1.b is misleading. You are suggesting architecture in Figure 2; reconciling it with Figure 1.b takes time.

**Other Strengths And Weaknesses:**

The key advantage of the proposed approach is the simplicity of zero-shot SPD and its apparent effectiveness.

The authors suggest B2B which includes distillation. HG includes modifynig the architecture. Both of these techniques are much more complex than zero-shot SPD and don’t seem to recover performance to a significant degree. The motivation for me to use them is unclear.

**Questions For Authors:**

The top concern that is not addressed by the paper:
1. How does SPD interplay and compare with other latency reduction techniques (quantization, pruning, block skipping)? Do I actually need SPD if I already have all of them?

1.a How much does sync-point dropping affect the communication-computation gap (the difference between the percentage of time comm channels are fully utilized and the percentage of time comp kernels are fully utilized)? One would expect that the first all-reduce in the block is a major bottleneck since most of the computation is actually happening in the feed-forward layer.

**Relation To Broader Scientific Literature:**

The Section 2 of the paper gives a good overview of the research on improving the latency of comms-bound inference systems. Quantization, pruning, and block-skipping reduce the communication load.

The paper only touches upon improving a single type of model parallelism, namely, the tensor parallelism. Other important types, the pipeline parallelism and the expert parallelism, are left aside.

The paper does not reflect on the other common types of bottlenecks that could impact latency: memory and computation bottlenecks.

**Theoretical Claims:**

The paper is empirical.

---

> ### Author Rebuttal · Authors · 2025-04-01
>
> We are deeply grateful for your support of our work, and we provide detailed responses to your comments as follows:
>
> **Q1. The paper would benefit from quantifying the potential speed-up in set-ups that do not suffer communication bottlenecks.**
>
> Thanks for the comment. We plan to add more details to the final draft.
>
> SPD targets the communication bottlenecks incurred by all-reduce operation after self attention. In Figure 7, we utilize 100% percentage of SPD block to total number of blocks as maximum speed-up case. In this case, all communications after self attention are eliminated while all communications at the end of the block (after feed forward network) are remaining. However, using the case of removing all the communications in model which doesn’t suffer from communication bottleneck is impossible since the proposed block structure and sensitivity measuring of SPD cannot be exploited in this case. In conclusion, instead of measuring latency of all communications removed cases which is not realizable, we used latency from removing all communications after self attention which is the best case in realizable solution.
>
> **Q2. The way sensitivity is measured looks arbitrary. Why don’t I measure sensitivity by applying SPD to individual blocks or applying SPD to the blocks at the beginning of the model?**
>
> Thank you for the question. When measuring the sensitivity of a block to SPD, we apply SPD to consecutive blocks starting from the selected block and extending toward the last block. This approach allows us to evaluate the worst-case impact of SPD on the block’s output while ensuring that its input activations remain numerically identical to the original, unaffected by prior SPD modifications. By analyzing the perplexity difference between a given block and the next block, we can isolate the effect of SPD at that specific layer, excluding any influence from subsequent layers. This method provides a fast yet precise and independent measurement of each block's sensitivity.
> The sensitivity analysis reveals that the impact of SPD varies significantly across different layers. For instance, in Qwen2.5-7B ([Reviewer USNW - Q2](https://openreview.net/forum?id=23zxLtvder¬eId=ZTV6Aoih0B)), the sensitivity ranking of blocks from least to most sensitive is observed as (26-24-5-25-22-17-11-20-21-18-12-10-7-14-16-15-4-23-19-8-2-13-9-3-6-1-27-0). This highlights that some layers are highly resilient to SPD, whereas others are more sensitive, underscoring the importance of a structured approach in determining which blocks to apply SPD while minimizing its impact on model accuracy.
>
> **Q3. The authors suggest B2B distillation. HG includes modifynig the architecture. The motivation for me to use them is unclear.**
>
> Thank you for the comment. While B2B and HG may not provide a perfect accuracy without loss, they offer valuable options for users seeking more aggressive optimizations. Additionally, HG does not modify the model architecture but rather permutes weights before distillation, consequently having same shapes of parameters. This simple adjustment helps improve accuracy after distillation in a distributed setting, making it a useful technique for enhancing performance without structural changes to the model.
>
> **Q4. Figure 1.b is misleading. You are suggesting architecture in Figure 2.**
>
> Thank you for the comment. Figure 1.b is a conceptual illustration showing the removal of sync operations, while Figure 2 presents the proposed block structure designed to achieve this without accuracy loss. To clarify the distinction, we will revise the title of Figure 1.b to "With elimination of sync-point in attention output", emphasizing that it does not represent SPD itself but rather illustrates the conceptual effect of removing sync points.
>
> **Q5.  How does SPD interplay with other latency reduction techniques (quantization, pruning, block skipping)? Do I actually need SPD if I already have all of them?**
>
> SPD is an orthogonal approach focusing on communication efficiency to other LLM optimizations targeting computation efficiency, e.g. efficient attention, quantization, pruning. This means that while computation has improved, communication is not optimized, making synchronization a larger bottleneck than before. This leads to a greater proportion of time spent in end-to-end model execution. In this circumstance, SPD can addresses this issue by reducing the number of sync points rather than just making them faster.
>
> **Q5.a. How much does sync-point dropping affect the communication-computation gap? One would expect that the first all-reduce in the block is a major bottleneck.**
>
> Since SPD is optimizing communication channel, after SPD, the percentage of communication time channel is reduced. However, the difference between first all-reduce (after self attention) and second all-reduce (after feed-forward layer) are negligible since they use same tensor size for communication and they cannot be overlapped in both cases.

---

### Official Review · Reviewer_USNW · 2025-03-09

**Overall Recommendation:** 3

**Summary:**

This paper introduces SPD, a novel optimization to reduce communication overheads in tensor parallelism b selectively dropping synchronization on attention outputs. SPD categorizes attention blocks into 3 types based on their sensitivity to the model accuracy, and applies different block design. SDP effectively reduce the communication while minimizing accuracy degradation.

**Claims And Evidence:**

Claims are supported by clear and convincing evidence.

**Essential References Not Discussed:**

No.

**Experimental Designs Or Analyses:**

The experimental design is sound.

**Methods And Evaluation Criteria:**

Methods and evaluation make sense.

**Other Comments Or Suggestions:**

Thanks for submitting to ICML '25!

**Other Strengths And Weaknesses:**

Strengths
1. The paper presents a novel solution to alleviate the communication bottleneck for LLM inference with tensor parallelism.
2. SDP achieves good speedup without accuracy degradation.

Weaknesses
1. Insufficient details about how to set hyper-parameters $\tau_1$ and $\tau_2$
2. It remains unclear how well the method generalizes to modern LLM architectures featuring GQA and MoE.

**Questions For Authors:**

1. How does the method affect accuracy on tasks that require handling long context lengths?
2. Is it possible to integrate this approach with sparsification techniques such as sparse attention?
3. These zero-shot tasks evaluated typically generate a single token. What is the impact on generation accuracy when the model is required to produce a longer sequence?

**Relation To Broader Scientific Literature:**

This paper helps improve the LLM inference efficiency when utilizing tensor parallelism.

**Theoretical Claims:**

NA

---

> ### Author Rebuttal · Authors · 2025-04-01
>
> Thanks for your support on our work:
>
> **Q1. Insufficient details about how to set hyper-parameters**
>
> Thank you for the comment. While we will add more details on hyper-parameters to the final draft, we like to point out one observation we made during our study about τ1 and τ2. We observed that there are distinct points where the sensitivity metric (perplexity difference with consecutive SPD blocks) sharply increases as SPD is applied to more sensitive blocks. These points served as natural candidates for setting τ1 (over 0.05 perplexity difference) and τ2 (over 10 perplexity difference), allowing us to categorize blocks into insensitive, sensitive, and extremely sensitive groups. This approach strikes a balance between maintaining model accuracy and achieving communication reduction, as described in Section 4.2. The results in Section 5 show that the thresholds are robust across different configurations, minimizing accuracy degradation while maximizing the benefits of SPD. We will add this details in the final version of paper.
>
> **Q2. It remains unclear how well the method generalizes to modern LLM architectures featuring GQA and MoE.**
>
> We are happy to share more experimental results around the modern LLM architectures. In general, the blocks and algorithms of SPD can be applied to any model or system, as long as tensor parallelism (TP) can be applied.
>
> **1. GQA:**
>
> The each head of key and value can be firstly divided across GPUs and the matching queries can be followed as in TP. We provide more results of Qwen2.5-7B on zero-shot tasks (same as the presented results of Figure 7) in the Table below. The results show that SPD can be successfully applied to LLM architectures with GQA.
>
> Table: Qwen2.5-7B 4-GPUs distributed inference average accuracy on zero-shot tasks (ARC, HellaSwag, LAMBADA, PIQA, SciQ, WinoGrande). Notations are same as in Figure 7 (‘ZS’: applying zero-shot dropping. ‘ZS+B2B’: applying ZS on ISBs and block-to-block distillation to the other remaining SBs. ‘ZS+B2B+HG’: applying ZS on ISBs, B2B on SBs and B2B with head grouping initialization to the other remaining ESBs.)
> | SPD drop percentage | 0% | 25% | 50% | 64.3% | 75.0% | 85.7% | 92.9% | 96.4% | 100% |
> | --- | :---: | :---: | :---: | :---: | :---: | :---: | :---: | :---: | :---: |
> | ZS | 75.3 | 73.8 | 72.0 | 71.6 | 71.5 | 68.9 | 65.8 | 60.3 | 38.9 |
> | ZS+B2B | 75.3 |  | 72.8 | 72.6 | 72.7 | 70.4 | 70.5 | 64.6 | 61.2 |
> | ZS+B2B+HG | 75.3 |  |  |  |  |  |  |  | 61.6|
>
> **2. MoE:**
>
> Due to the space limit, we would appreciate if the reviewer can check out our response to [Reviewer e4AF - Q3](https://openreview.net/forum?id=23zxLtvder¬eId=q5CWzbGn8w).
>
> **Q3. Is it possible to integrate this approach with sparsification techniques such as sparse attention?**
>
> Thank you for the question, and the our answer is yes.
> SPD is an orthogonal approach focusing on communication efficiency to other LLM optimizations, e.g. sparse attention, quantization, pruning. Sparse attention can be easily integrated into distributed attention weights distributed in each devices. The distributed attention weights are derived from query and key which are came from partial linear Q, K projection layers divided along the head dimension. Therefore, diverse sparse attention techniques working within each head dimension of partial attention weights are easily applicable with SPD settings.
>
> **Q4. How does the method affect accuracy on tasks that require handling long context lengths? + Q5. These zero-shot tasks evaluated typically generate a single token. What is the impact on generation accuracy when the model is required to produce a longer sequence?**
>
> Thank you for raising important question and we conducted extra experiments to find answer to the question.
> The Table below shows Longbench-Multinews (summarization) benchmark [1] on Llama-2-7b-chat. Similar with Figure 7 and 8 of the paper, in generation task, SPD effectively alleviate communication bottlenecks while minimizing accuracy degradation during LLM inference.
>
> Table: Longbench-Multinews Rouge-L score on Llama-2-7b-chat 8-GPUs distributed
> | SPD drop percentage | 0% | 25% | 50% | 62.5% | 75.0% | 84.4% | 93.8% | 96.9% | 100% |
> | --- | :---: | :---: | :---: | :---: | :---: | :---: | :---: | :---: | :---: |
> | ZS | 25.8 | 26.0 | 25.6 | 25.4 | 25.2 | 18.3 | 7.1 | 4.2 | 1.2 |
> | ZS+B2B | 25.8 |  |  | 25.4 | 25.4 | 25.1 | 24.9 | 24.3 | 20.1 |
> | ZS+B2B+HG | 25.8 |  |  |  |  |  |  | 24.7 | 22.7 |
>
> [1] https://github.com/THUDM/LongBench/tree/main/LongBench#summarization
>
> [2] Liu et al, Deja Vu: Contextual Sparsity for Efficient LLMs at Inference Time, ICML 2023
>
> [3] Agarwal et al, CHAI: Clustered Head Attention for Efficient LLM Inference, ICML 2024
>
> [4] Agrawal et al, exmy: A data type and technique for arbitrary bit precision quantization, 2024
>
> [5] Dong et al, Towards Low-bit Communication for Tensor Parallel LLM Inference, NeurIPS 2024
>
> [6] Nvidia, https://developer.nvidia.com/blog/nvidia-hopper-architecture-in-depth/

---

### Official Review · Reviewer_LBEy · 2025-03-11

**Overall Recommendation:** 3

**Summary:**

This paper proposes a novel method to drop one of the two all-reduce communications in tensor parallelism (TP) to mitigate the communication of TP in LLM inference. After dropping the first all-reduce communication, the sync point between the attention and mlp is removed, raising concerns on the model accuracy. To reduce the negative impact on accuracy of the model, the paper proposed several methods:
- Modifications to model architecture, especially residual links to retain most accurate semantics
- Classify the model blocks (transformer layers) into insensitive, sensitive and extremely sensitive blocks by the impact it would have if the sync point in this layer is dropped.
- For sensitive and extremely sensitive layers, the authors proposed several methods including distillation and to recover the performance loss
The experiments results shows 10%-20% performance improvements with variant performance loss. In the best case where the communication cost is high due to low communication bandwidth, the method can achieve 20% acceleration with less than 1% accuracy loss.

**Claims And Evidence:**

One of the key insight of this paper lies in the finding that most (>50%) layers can drop the TP sync point without significant latency, which is well supported by the experiments, where <= 50% zero shot drop usually have negligible accuracy loss.

However, the performance gain is a bit overclaimed if conditioned on negligible accuracy loss. It is misleading to claim 20% acceleration. According to Figure 7 (a), (b) (c), which is the most common setting to use TP on 8 GPUs on llama2 models, to keep <1% performance achieves approximately 5% latency gain if NVLINK is used. There is a 20% latency improvement under the condition of running 70B model on 8 GPUs with only a bandwith of only 10GB, which is not a common setting for running TP. Notice that SPD cuts the communication volume by at most 50%.

This paper claims to improve the scalability of TP, but no direct scaling evaluation is shown. For example, how does the improved latency changes if the number of GPUs scales out?

Moreover, there is a divergence between the performance of OPT and llama 2 models, suggesting the method might have different performance in terms of accuracy loss on different models.

**Essential References Not Discussed:**

N/A

**Experimental Designs Or Analyses:**

10 GB bandwidth is too low for an efficient deployment of TP. If the paper wants to claim TP becomes practical in this scenarios, it should compare with other model parallelism schemes applicable in model inference. At least, the paper should reveal more common metrics of a model inference system like throughput, time between tokens (TBT), etc so readers can get an idea whether the baseline is a legit baseline.

Overall, for HBW senarios, the latency improvement is around 5% if the accuracy loss is negligible. The performance gain in practical cases seem to be moderate.

**Methods And Evaluation Criteria:**

The SPD method and the accuracy recovery methods for SB and ESB blocks are well designed. Their model accuracy is verified by evaluations properly.
This paper compares the latency with baseline methods. It is unclear what this latency exactly means. e.g. is it e2e latency of a single request? or an iteration in LLM inference?

**Other Comments Or Suggestions:**

The header of each page is still 'Submission and Formatting Instructions'.

**Other Strengths And Weaknesses:**

Strengths: An effective method showing dropping sync points in TP might be beneficial with negligible accuracy loss
Other Weakness:
- Lack of impact: From the evaluations, a safe threshold for guaranteeing a negligible accuracy loss is around cutting 50% sync points, which results in 25% reduction of the overall TP communication volume.

**Questions For Authors:**

- What is the definition of latency in the evaluations?
- Why the SPD blocks have to be consecutive blocks from the last block?

**Relation To Broader Scientific Literature:**

The method introduced in this paper would accelerate LLM serving systems.

**Theoretical Claims:**

The methods involved in model accuracy recovery are well supported by empirical evaluations.

---

> ### Author Rebuttal · Authors · 2025-04-01
>
> **Q1. This paper claims to improve the scalability of TP, but no direct scaling evaluation is shown.**
>
> Thank you for the comment. While we haven’t directly measured the scalability of TP with SPD, we can intuitively understand that SPD would lead to better scaling efficiency as GPU counts increase since SPD surgically targets the communication overhead (which is a key barrier to high quality scalability). The question is how much better and it also depends on many complex factors including the compute/communication ratio as discussed with [Reviewer e4AF - Q4](https://openreview.net/forum?id=23zxLtvder¬eId=q5CWzbGn8w).
> We couldn’t directly measure the scalability due to the time and resource limit during this rebuttal period, but plan to add detailed scalability measurements in the final draft.
>
> **Q2. The method might have different performance on different models.**
>
> Thank you for the important comments. For more data point, we experimented with Qwen2.5-7B [Reviewer USNW - Q2](https://openreview.net/forum?id=23zxLtvder¬eId=ZTV6Aoih0B) and Longbench-Multinews (summarization) on [Reviewer USNW - Q4](https://openreview.net/forum?id=23zxLtvder¬eId=ZTV6Aoih0B). In short, our new results indicate SPD shows the comparable effectives as it showed on the paper.
>
> **Q3. Lack of impact: A safe threshold for a negligible accuracy loss is 50% SPD results in 25% reduction of the overall TP volume.**
>
> Thank you for the chance to discuss the impacts. Although a 25% reduction may seem modest at first, achieving this reduction with no additional cost is significant. Most model-level optimizations, such as quantization or sparsity, require preprocessing steps or specialized hardware to realize performance gains. In contrast, SPD selectively removes sync point for blocks by simply not performing all-reduce, effectively removing 25% with negligible accuracy loss. Furthermore, in larger model (Llama2-70B), 70% of SPD results in 35% reduction. These results demonstrate that SPD offers a very low cost approach to improve efficiency in large-scale distributed inference. Also, note that SPD provides a scalable communication that adapts to different budget constraints while minimizing accuracy loss.
>
> **Q4. What is the definition of latency in the evaluations?**
>
> Thank you for the chance for clarification. The latency reported is end-to-end model latency when generating one token on prompting (time to first token) with batch size 1 and prompt length 128 on A100 by following the metrics utilized in existing optimizations [2,3] . On Figure 2, we will add the definition of latency. On Figure 7, to provide the intuitive comparison of reduced latency, we displayed the speedup with improved latency of SPD from latency without SPD. For the decoding latency, since the operations are bottlenecked to memory, the communication optimization including SPD and existing works [4,5] does not give great improvement compared to prefill stage. We will add more details to the final version.
>
> **Q5-1. 10GB bandwidth is too low.**
>
> Since the GPU only supports interconnect with NVLink (300GB/s) and without NVLink (10GB/s), there are no more selections that we can test with.
>
> **Q5-2. For high-bandwidth (HBW) scenarios, the latency improvement is around 5% without accuracy loss. The performance gain in practical case seem to be moderate.**
>
> General-purpose infrastructure, such as GPU servers, is designed for high performance across all aspects, featuring hardware support that minimizes communication overhead. As a result, with HBW, the latency improvement from SPD may appear moderate. However, in specialized infrastructures, particularly inference-focused hardware, cost constraints often limit HBW interconnect support compared to GPUs. In such environments, where communication bandwidth is a more significant bottleneck, SPD can play a crucial role in reducing sync overhead and improving overall efficiency. This makes SPD particularly valuable for inference infrastructure with limited interconnect performance.
>
> **Q5-3. If the paper wants to claim TP becomes practical in this scenarios, it should compare with other parallelisms.**
>
> Thank you for the comments. We design SPD for TP since TP is the most widely used techniques in distributed inference. Therefore, we focus our evaluation on TP to specifically address its communication bottleneck. Also, TP would work with other parallelisms in a hybrid mode (please see [Reviewer e4AF - Q3](https://openreview.net/forum?id=23zxLtvder¬eId=q5CWzbGn8w)). We plan to explore the suggestion as a future direction.
>
> **Q6. Why the SPD blocks have to be consecutive blocks from the last block?**
>
> Please refer to [Reviewer J4A1 - Q2](https://openreview.net/forum?id=23zxLtvder¬eId=ekQlcDOwVR). In short, measuring SPD sensitivity with consecutive blocks enables isolating a block’s impact by analyzing perplexity difference. We will add more details to the final version.
>
> \
> Reference: Please refer to the references of Reviewer USNW.

---

> > ### Comment · Reviewer_LBEy · 2025-04-03
> >
> > Thank you for your detailed response. However, I remain concerned about the potential overstatement of the performance improvement. While I acknowledge that the proposed method can be beneficial in low-bandwidth (LBW) settings, I am not convinced that these specific LBW conditions—even after SPD optimization—are practical for tensor parallelism (TP). In other words, are we evaluating the method and baselines in a scenario where both are inherently unsuitable and underperforming?
> >
> > Regarding your response to reviewer e4AF, you noted that communication accounts for 15% of execution time with NVLINK on an A100 GPU. Given that LBW settings have a bandwidth that is 30× lower (based on your data comparing 300GB/s NVLINK vs. 10GB/s PCIe), I estimate that communication would constitute approximately 84% of the total execution time:
> >
> >  $1-0.85 / (0.15 * 30 + 0.85) = 0.84$
> >
> > Even with SPD reducing communication by 25%, the communication overhead would still be around 80%:
> >
> > $1-0.85 / (0.15 * 30 * 0.75 + 0.85) = 0.8$
> >
> > This suggests that in such scenarios, TP may not be the optimal approach. While TP is commonly used due to the assumption that NVLINK is available in most inference systems, it is not the only viable option. Pipeline parallelism (PP), which involves significantly lower communication overhead, may be a more suitable alternative in this case.

---

> > > ### Author Response · Authors · 2025-04-07
> > >
> > > Thank you for your thoughtful follow-up and the constructive example provided.
> > >
> > > We agree that the optimal inference parallelism strategy is highly context-dependent, influenced by factors such as the compute-to-communication ratio and their respective hardware efficiencies. While Tensor Parallelism (TP) is not universally superior to Pipeline Parallelism (PP), it has been widely adopted in real-world LLM inference deployments—even in some bandwidth-constrained environments—due to its engineering simplicity and, more importantly, its superior compute utilization.
> > >
> > > This is particularly evident in the prefill stage of autoregressive models, where PP struggles with pipeline bubbles and complex workload balancing, leading to underutilized compute resources. In contrast, TP delivers more consistent and higher utilization across devices, making it the preferred choice in many production settings.
> > > We appreciate the reviewer’s example (summarized in the table below) as a valuable case study to further this discussion.
> > >
> > > | |Computation Latency|Communication Latency|Computation (%)|Communication (%)|Total Latency|Speedup from LBW TP|
> > > |---|:---:|:---:|:---:|:---:|:---:|:---:|
> > > |HBW TP|0.85|0.15|85%|15%|1|n/a|
> > > |LBW TP (by reviewer)|0.85|4.5|15.89%|84.11%|5.35|1.0|
> > > |LBW TP+SPD (by reviewer)|0.85|3.375|20.12%|79.88%|4.23|1.27|
> > > |LBW PP (by authors)|6.8(0.85x8)|0.2|97.19%|2.81%|7.0|0.76|
> > >
> > > **4% Reduction in Communication Bottleneck**
> > >
> > > In the Low-BW TP setup, the reviewer correctly observes that SPD reduces the communication portion by 4%. However, we would like to emphasize that even this modest reduction can translate into a significant 27% improvement in end-to-end latency—a metric that directly impacts user-perceived performance.
> > >
> > > This discrepancy arises because once the communication bottleneck is alleviated, the remaining latency becomes increasingly dominated by computation. Hence, the communication portion appears smaller in relative terms, even though the absolute latency reduction is substantial. In that sense, the 4% figure underrepresents the true impact of SPD on overall latency.
> > > Therefore, we believe this result underscores SPD's effectiveness in accelerating inference, particularly in bandwidth-constrained scenarios, as discussed in our paper.
> > >
> > > **TP vs PP**
> > >
> > > To further contextualize the tradeoffs between TP and PP, we conducted an apples-to-apples comparison using the reviewer’s example, as shown in the last row of the table (we did paper-pencil study of the 70B to saturate the 8 GPU cluster).
> > >
> > > * We first deploy LLaMA2-70B using an 8-GPU TP configuration and determine the maximum supported prefill length (approximately 1632 tokens).
> > > * We then replicate this maximum prefill length using a PP configuration of LLaMA2-70B, adjusting the number of pipeline stages accordingly to find the minimum viable number of stages. On our system, this results in 8 stages (i.e., 8 GPUs).
> > >
> > > In this setup, GPU memory is fully utilized, allowing only a single user to be served at a time, which is also a common deployment constraint in industrial environments where strict privacy guarantees are enforced. Under such conditions, PP suffers from substantial GPU underutilization, as only one pipeline stage (i.e., one GPU) is active at any given time during prefill. This leads to up to an 8× increase in compute latency (the total FLOPs required for prefill don't change by PP vs TP), as demonstrated in our measurements. While PP may offer good improvements in communication efficiency, these are far outweighed by low compute utilization, ultimately resulting in worse overall latency compared to the TP baseline—the metric that most directly impacts user experience.
> > >
> > > While prior work has explored more sophisticated strategies (e.g., mixing prefill and decode stages across multiple users [7]), these approaches introduce considerable engineering complexity and still suffer from pipeline bubbles and scheduling inefficiencies, especially under dynamic or low-throughput workloads. For these reasons, TP remains the preferred solution in many real-world inference deployments, striking a practical balance between performance and system complexity.
> > >
> > > **Final remark**
> > >
> > > Yet, more broadly, we acknowledge the reviewer’s valid point that PP can be more advantageous than TP in certain scenarios, depending on system workload, user concurrency, and hardware architecture. While our work focuses on improving TP in isolation, prior work [8] has explored hybrid strategies combining TP and PP, which can yield better trade-offs under particular conditions. We agree that a more comprehensive evaluation—including alternative parallelism strategies—could provide additional valuable insights. We appreciate this suggestion and plan to incorporate a broader analysis in future work. In the meantime, we will ensure that our performance claims are clearly contextualized and scoped in the final version.
> > >
> > > [7] https://arxiv.org/abs/2308.16369
> > >
> > > [8] https://arxiv.org/abs/2104.04473

---

### Official Review · Reviewer_e4AF · 2025-03-13

**Overall Recommendation:** 3

**Summary:**

This paper introduces SPD aiming at reducing communication overhead in tensor parallelism for LLMs. SPD selectively removes synchronization points in attention outputs to mitigate latency bottlenecks during distributed inference. The authors propose a block-based SPD approach and classify transformer blocks into three sensitivity categories (insensitive, sensitive, and extremely sensitive) to balance performance trade-offs. Their empirical results on LLaMA2 and OPT models demonstrate up to 20% inference speedup with less than 1% accuracy regression.

**Claims And Evidence:**

1. SPD reduces inference latency by selectively dropping synchronization in tensor parallel models.

    Supported: Experimental results on LLaMA2 and OPT models show significant speedup (up to 20%) with low accuracy degradation.

2. The block-wise sensitivity identification approach effectively balances performance and accuracy.

    Supported: Results indicate that zero-shot SPD can be applied to ~44-84% of blocks with minimal impact on accuracy.

**Essential References Not Discussed:**

There are no specific references currently on my mind that have not been discussed.

**Experimental Designs Or Analyses:**

The experiments effectively validate SPD’s performance:
1. Multiple LLM scales (7B-70B) and architectures (LLaMA2, OPT).
2. Both high-bandwidth and low-bandwidth scenarios.

However, there are also some concerns:
1. The evaluation is limited to LLaMA2 and OPT, which are relatively old.
2. The evaluation benchmarks are logprob-based and relatively simple. It would be appreciated if more generation based or reasoning tasks could be included.

**Methods And Evaluation Criteria:**

The paper employs:

1. Distributed inference on LLaMA2 (7B, 13B, 70B) and OPT (6.7B, 13B, 30B, 66B) across 4-GPU and 8-GPU setups.
2. Zero-shot accuracy tasks (ARC, HellaSwag, LAMBADA, etc.) and MMLU benchmarks to measure accuracy degradation.
3. Latency measurements across different interconnect bandwidths (300GB/s vs. 10GB/s).

Potential concerns:

1. The evaluation is limited to LLaMA2 and OPT models. Testing on more recent models (e.g., Llama 3.3, Qwen 2.5, DeepSeek V3/R1) could validate its generalizability.
2. The evaluation benchmarks are logprob-based tasks. This do not aligns with the goal of reducing decoding latency in this paper. It would be appreciated if more generation based or reasoning tasks will be included.

**Other Comments Or Suggestions:**

1. Would SPD work on MoE architectures or hybrid parallelism setups?
2. For recent hardware e.g. H200/H100, the communication speed is actually very fast. The gain of removing an AR may be incremental.

**Other Strengths And Weaknesses:**

Pros:
1. SPD provides a simple yet effective method by removing sync point to reduce the inference latency.
2. The latency vs. accuracy trade-offs are well-analyzed.

Cons:
1. The evaluated models are relatively outdated and weak. It would be great to evaluate on more recent LLMs.
2. The evaluation benchmarks are logprob-based and relatively simple. It would be appreciated if more generation based or reasoning tasks could be included.

**Questions For Authors:**

Please check "Other Comments Or Suggestions".

**Relation To Broader Scientific Literature:**

The paper builds on prior work in Tensor Parallelism, where SPD modifies this by selectively removing synchronization. There are other LLM Optimization, e.g. Quantization, Pruning, while SPD is an orthogonal approach that focuses on communication efficiency.

**Theoretical Claims:**

NA

---

> ### Author Rebuttal · Authors · 2025-04-01
>
> Thanks for your thoughtful advices to our work. Here are the replies to your concerns.
>
> **Q1. The evaluation is limited to LLaMA2 and OPT models. Testing on more recent models could validate its generalizability.**
>
> Due to space limit, we couldn’t share our new results here. Please refer to the displayed result of Qwen2.5-7B on [Reviewer USNW - Q2](https://openreview.net/forum?id=23zxLtvder¬eId=ZTV6Aoih0B).
>
> In short, we’ve observed comparable effectiveness of SPD on Qwen2.5-7B, further validating its generalizability to recent models. Thank you for your understanding.
>
> **Q2. The evaluation benchmarks are logprob-based tasks. It would be appreciated if more generation based or reasoning tasks will be included.**
>
> Again, due to space limit, we would appreciate if you can check out the displayed result of Longbench-Multinews (summarization) benchmark on Llama-2-7b-chat on [Reviewer USNW - Q4](https://openreview.net/forum?id=23zxLtvder¬eId=ZTV6Aoih0B).
>
> In short, we’ve obtained positive results on generation tasks with SPD, showing performance comparable to the logprob-based benchmarks.
>
> **Q3. Would SPD work on MoE architectures or hybrid parallelism setups?**
>
> Thank you for the chance to elaborate how SPD works with MoE and hybrid parallelisms. The blocks and algorithms of SPD can be applied to any model or system that tensor parallelism (TP) can be applied.
>
> * Applying on MoE architectures
>     * Token Routing (Top-k Experts Selection)
>         * In SPD, the routing network (softmax over token scores) can be replicated across GPUs following TP. The overhead with the replicated routers is negligible since they have much fewer parameters and require much less computation than the expert network. The router in each GPU assigns each token to the top-k selected experts having same decision per devices. GPUs begin expert computations immediately with local routing decisions.
>     * Expert Feedforward Layers (Sparse Computation)
>         * Each expert is a fully connected feedforward network (FFN). The selected experts process only their assigned tokens, with evenly distributed to all GPUs. After computation, the expert outputs are processed through next FFN layer, then requiring final all-reduce communication between GPUs.
>     * Residual connection and bias in SPD block
>         * As in dense models, skip connections and bias connections can be applied using the same structure presented in Figure 3.
> * Applying on hybrid parallelism setup
>     * Data parallelism
>         * Copying a SPD model replica in GPU setting can make data parallelism work with SPD. For example, when we have 8-GPUs, we can set 4-GPUs distributed SPD model as one replica and set the copied replica to the other 4-GPUs so that 2 replicas are running as data parallelism setup.
>     * Pipeline parallelism
>         * Pipelining each parallelized execution of SPD (a fraction of attention and FFN of one block in a single GPU branch) can make pipeline parallelism work with SPD. For example, we have 4-GPUs distributed SPD block. In this case, we can divide a fraction of attention operation and FFN operation of one GPU to a pair of GPUs. As a result, 4 x 2 - GPUs server can work for pipeline parallelism setup and SPD eliminates all-reduce operation of former GPU in each 4-GPU-pairs.
>     * Combined data and pipeline parallelism
>         * One pipeline parallelized model can be set as a replica for data parallelism and the copied replica can make data parallelism setup with pipeline parallelized setup. This can finally make hybrid parallelism working with SPD ({TP+SPD} + data parallelism + pipeline parallelism)
>
> **Q4. For recent hardware e.g. H200/H100, the communication speed is actually very fast. The gain of removing an AR may be incremental.**
>
> Thank you for your comments. We first like to share how H100 is compared with A100 on computation and communication.[6]
>
> * H100 offers 3.17× higher computation performance (on FP16 TFLOPS)
> * H100 offers 1.5× faster NVLink interconnect for device-to-device communication
>
> Hence, we can easily say that overall inference latency will go down thanks to stronger HW capability. However, we’d like to highlight that since compute has grown even faster, communication becomes an even larger bottleneck than before. In the Table below, we projected how compute vs. communication ratio changes for Llama2-7B 8-GPUs distributed inference when we move from A100 to H100, and it clearly shows that now the communication takes about 1.8x larger portion of the overall latency (i.e. from 15% to 27%), thus making SPD would be even more important. We hope this address your concern around how SPD will help with the recent hardware advancements.
>
> Table: H100 projected computation/communication ratio compared to A100 on HBW (300GB/s) interconnect (Llama2-7B 8-GPUs)
> |  | Computation (FP16) | Communication |
> | --- | :---: | :---: |
> | A100 | 85% | 15% |
> | H100 (projected) | 72.8% | 27.2% |
>
> \
> Reference: Please refer to the references of Reviewer USNW.

---

### Decision · Program_Chairs · 2025-05-01

**Decision:**

Accept (poster)

**Comment:**

This paper introduces Sync-Point Drop (SPD), a novel technique to mitigate communication bottlenecks in distributed LLM inference by selectively skipping synchronization steps in attention blocks during tensor parallelism. SPD’s block design enables communication-free execution while balancing accuracy and efficiency, achieving a 20% reduction in inference latency with <1% accuracy loss for LLaMA2-70B on 8 GPUs.

During rebuttal, the authors effectively addressed reviewer concerns, including SPD’s applicability to modern accelerators, improved user-perceived latency in bandwidth-constrained scenarios, and its advantages over pipeline parallelism in some scenarios. These clarifications led to unanimous positive ratings from reviewers. The AC also concurs with the recommendation to accept this paper.